# Selection Performance Using a Scaled Virtual Stylus Cursor in VR

Seyed Amir Ahmad Didehkhorshid*

Carleton University, Ottawa, Canada

Robert J. Teather**

Carleton University, Ottawa, Canada

## ABSTRACT

We propose a surface warping technique we call warped virtual surfaces (WVS). WVS is similar to applying CD gain to mouse cursor on a screen and is used with traditionally 1:1 input devices, in our case, a tablet and stylus, for use with VR head-mounted displays (HMDs). WVS allows users to interact with arbitrarily large virtual panels in VR while getting the benefits of passive haptic feedback from a fixed-sized physical panel. To determine the extent to which WVS affects user performance, we conducted an experiment with 24 participants using a Fitts' law reciprocal tapping task to compare different scale factors. Results indicate there was a significant difference in movement time for large scale factors. However, for throughput (ranging from 3.35 - 3.47 bps) and error rate (ranging from 3.6 - 5.4%), our analysis did not find a significant difference between scale factors. Using non-inferiority statistical testing (a form of equivalence testing), we show that performance in terms of throughput and error rate for large scale factors is no worse than a 1-to-1 mapping. Our results suggest WVS is a promising way of providing large tactile surfaces in VR, using small physical surfaces, and with little impact on user performance.

**Index Terms:** • Human-centered computing~Human computer interaction (HCI)~Interaction techniques~Pointing • Human-centered computing~Human computer interaction (HCI)~Interaction paradigms~Virtual reality

## 1 INTRODUCTION

There has been a recent surge in demand for virtual reality (VR) in entertainment, education, and design applications. Current VR hardware is self-contained, wireless, lighter, and offers high visual fidelity. Development tools (e.g., Unity3D) are also becoming more accessible. These factors have created the perfect storm, paving the way for a new wave of innovations and creativity in immersive VR technologies. Despite these advances, there remain challenging research problems to be solved. General-purpose haptics is among these big problems. Past research has shown that haptic feedback significantly increases the quality of a VR experience [25, 31, 34, 51]. However, designing interaction techniques that support realistic haptic feedback in VR is still problematic.

Past studies have demonstrated the effectiveness of planar surfaces as a semi-general purpose prop in VR. In particular, the use of tablets to provide tactile surfaces in VR has been extensively studied [11, 18, 44, 53, 63, 68]. The Personal Interaction Panel (PIP) [63], Lindeman et al.'s HARP system [44], the Virtual Notepad by Poupyrev et al. [53], and Worlds In Miniature (WIM) [61] all used tracked panels for VR interaction. Other studies used a tablet and stylus for text input in VR [11].

Several researchers have investigated the use of redirection and retargeting techniques for VR interaction. This relatively new class

---

\* amir.didehkhorshid@carleton.ca

\*\* rob.teather@carleton.ca

Graphics Interface Conference 2020
28-29 May

of perceptual illusion-based interaction techniques include techniques like redirected walking (RDW) [55], haptic retargeting [6], and redirected touch [38]. Except for Yang et al.'s VRGrabber technique, which used retargeting with grabbing tools [71], all other techniques apply warping or redirection to the entire body (e.g., with RDW), or a body part such as the hand or fingers. None of the proposed interaction techniques so far have been applied with a planar surface, despite the well-known benefits of using these in VR. Although there are studies on bimanual retargeting [27, 50] or unimanual redirected touching that used planar surfaces in their experiments [38–40], none studied the planar surface as a means of input method. Large displays are known to offer performance benefits on spatial tasks, spatial knowledge, and navigation [7, 64, 65]. Hence, we propose to use space warping to extend the virtual interaction panel surface in VR.

We propose a technique we call Warped Virtual Surfaces (WVS). WVS combines the ideas behind RDW in expanding the tracking space, with haptic retargeting to use perceptual illusions and body warping. With WVS, users can interact with an arbitrarily large tactile virtual surface in HMD-based VR. Haptic feedback is provided by a fixed-size tracked physical tablet that uses a stylus for input. The technique applies a scale factor (SF) to move the virtual cursor beyond the tablet's active tracking area, making the stylus behave like an indirect input device, similar to a mouse and akin to changing the control-display (CD) gain [5, 48].

Since users cannot see their physical hands or the stylus when using HMDs, they do not notice the decoupling between the physical stylus and virtual cursor positions. In non-VR setups, this could be distracting and confusing. Different SFs cause the cursor to move further with less physical movement on the tablet, much like how CD gain works for a mouse cursor. However, the user perceives the virtual contact on the surface in VR and thus receives appropriate tactile cues from physically touching the tablet with the stylus. WVS creates the illusion of an arbitrarily large tactile surface in VR while keeping the users' motor space consistent during interaction on the tablet. WVS can theoretically be employed with any other tablet-based VR technique, offering tactile feedback with a virtually larger tablet than what is available.

We conducted an experiment to determine how far we can push this illusion and expand the interaction surface without affecting user performance. To evaluate user performance, we employed a Fitts' law reciprocal tapping task [24, 66]. To our knowledge, our experiment is the first to investigate the effects of surface warping using a stylus on user performance in selection tasks, and to apply CD gain with a tablet and stylus input modality in VR.

## 2 RELATED WORK

### 2.1 Visual Illusions

Studies on the human brain reveal the dominance of vision over other senses when there are sensory conflicts [9, 16, 21, 26, 28, 57, 60]. For instance, Gibson showed that a flat surface is perceived as curved while wearing distortion glasses and moving hands on a straight line [26]. VR and HCI researchers took advantage of this visual dominance to enhance selection in VR [4, 52, 54, 69], yielding several novel interaction techniques for HMD-based VR.

Many such techniques rely on the inaccuracy inherent to our proprioception and vestibular senses [9]. Burns et al. provided

Table 1. Summary of key studies on perceptual illusion techniques in VR.

| 1st Author/Year | Description | Evaluation Method | Redirection on | Main Findings | Notes |
|---|---|---|---|---|---|
| Kohli/2010 [38, 39, 40] | Redirected touch under warped spaces. | Fitts' law | Fingers | Discrepant conditions are no worse than 1-to-1 mapping. | Also evaluated performance & adaptation under warped spaces. |
| Azmandian/2016 [6] | Repurposing a passive haptic prop using body, world & hybrid space warping. | Block stacking | Hands | Hybrid warping scored the highest presence score. | Recommended encouraging slow hand movements & taking advantage of visual dominance. |
| Carvalheiro/2016 [16] | Haptic interaction system for VR & evaluating user awareness regarding the space warping. | Touch & moving objects | Hands | Average accuracy error of 7mm, users adapt to distortion & are less sensitive to negative distortions. | 7 participants did not detect distortions, even though they were told about it in the second phase of experiments. |
| Murillo/2017 [52] | Multi-object retargeting optimized for upper limb interactions in VR. | Target selection | Hands | Improved ergonomics with no loss of performance or sense of control. | Used tetrahedrons to partition the physical and virtual world for multi-object retargeting. |
| Cheng/2017 [19] | Sparse haptic proxy using gaze and hand movement for target prediction. | Target acquisition | Hands | Retargeting of up to 15° similar to no retargeting, 45° received lower but above natural ratings. | Predicted desired targets 2 seconds before participants touched with 97.5% accuracy. |
| Han/2018 [28] | Static translation offsets vs. dynamic interpolations for redirected reach. | Reaching for objects | Hands | The translational technique performed better, with more robustness in larger mismatches. | Horizontal offsets up to 76cm applied when reaching for a virtual target were tolerable. |
| Yang/2018 [71] | Virtual grabbing tool with ungrounded haptic retargeting. | Using controller for precise object grabbing | Controller | Travelling distance difference between the visual and the physical chopstick needs to be in the range (−1.48,1.95) cm. | Control/Display ratio needs to be between 0.71 and 1.77 & better performance with ungrounded haptic retargeting. |
| Feuchtner/2018 [23] | Slow shift of user's virtual hand to reduce strain of in-air interaction. | Pursuit tracking | Hands | Vertical shift hand by 65cm reduced fatigue, maintained body ownership. | Vertical shift decreases performance by 4% & gradual shifts are preferable. |
| Matthews/2019 [50] | Bimanual haptic retargeting with interface, body & combined warps. | Pressing virtual buttons | Hands/ Bimanual | Faster response time for combined warp. increased error in body warp. | Same time and error between bimanual and unimanual retargeting, but needs a more statistically powerful study. |

evidence of visual dominance over proprioception in a series of studies and found that people tend to believe their hand is where they see it [13–15]. Klatzky et al. showed that vestibular cues are also dominated by vision [37]. These ideas have been applied in various HCI contexts. For example, Zenner et al. used an internal weight shifting mechanism in a passive haptic proxy to enhance virtual object length and thickness perception with Shifty [73]. Similarly, Krekhov et al. used weight perception illusions in a self-transforming controller to enhance VR player experience [42]. McClelland et al. introduced the Haptobend, which used a bendable device to support different objects with simple geometry such as tubes and flat surfaces with a single physical prop [51].

Vision is not always dominant. In the case of conflicts, sensory signals are weighted based on reliability in the brain [30, 32]. There are thresholds on the dominance of vision. Some studies used the just noticeable difference (JND) threshold methodology to quantify mismatch thresholds [13, 36, 43, 49] while others employed two-alternative forced-choice (2AFC) [72]. Interestingly, some studies have shown that force direction and the curvature of real props can influence the mismatch thresholds [8, 56, 72].

## 2.2 Perceptual Illusions in VR

Table 1 summarizes key studies on perceptual illusions in VR. Haptic retargeting, introduced by Azmandian et al. [6], partially solved a major limitation of using physical props for tactile feedback, by mapping one physical object to multiple virtual ones. The technique operates by redirecting the user's hand towards the physical prop when they are reaching for different virtual items at various locations [6]. This and similar techniques work through perceptual illusions and the dominance of vision over other senses [9, 26, 37, 49, 57, 60, 62]. Likely the most well-known example is redirected walking, first proposed by Razzaque et al. [55]. RDW enables users to walk an infinite straight virtual space in HMD VR. In reality, RDW users are walking in circles in a limited tracking space, but perceive themselves as walking on straight lines.

Kohli et al. were among the first to propose redirected touching in a VR setting [38]. In a series of experiments, Kohli et al. looked into the effects of warping virtual spaces on user performance and adaptation and training under warped spaces [39, 40]. They reported that while training under real conditions seemed more productive, after adapting to discrepancies between vision and proprioception, participants performed much better [40]. Indeed, they report that participants had to readapt to the real world after adapting to the warping virtual space [40].

Azmandian et al. took the idea of redirected touching further and introduced haptic retargeting, which added dynamic mapping of the whole hand rather than just the fingers [6]. The technique leverages visual dominance to repurpose a single passive haptic prop for various virtual objects. This produced a higher sense of presence among participants, in line with past findings on the benefits of haptics in VR [34]. Their technique is limited by the shape of the physical prop, and that the target position must be known prior to selection. To overcome the targeting limitations, Murillo et al. proposed a multi-object retargeting technique by partitioning both virtual and physical spaces using tetrahedrons to allow open-ended hand movements while retargeting [52]. Haptic retargeting could also be applied for bimanual interactions [50]. Matthews et al. suggested that the technique could also be applied to wearables interfaces, i.e., on the user's wrist or arm [50].

Several other studies employed similar techniques. For example, Cheng et al. explored the applications and the limits of hand redirection using geometric primitives with touch feedback in a VE while predicting the desired targets using hand movements and gaze direction [19]. Feuchtner et al. proposed the Ownershift interaction technique to ease over the head interaction in VR while wearing an HMD [23]. Ownershift does not require a mental recalibration phase since the initial 1:1 mapping allowed initial ballistic movements toward the targets [23]. Abtahi et al. utilized Visuo-haptic illusions in tandem with shape displays [1]. They were able to increase the perceived resolution of the shape displays for a VR user by applying scales less than 1.8x, redirecting sloped lines with angles less than 40 degrees onto a horizontal line.

To summarize, despite the well-known advantages of large displays and planar surfaces in VR, very few studies have used warping techniques with planar input devices. Our proposed technique and present study aim to fill this gap.

## 2.3 Fitts' Law and Scale

Fitts' law predicts selection time as a function of target size and distance [24]. The model is given as:

$$MT = a + b \cdot ID \quad where \quad ID = log_2\left(\frac{A}{w} + 1\right) \quad (3)$$

where $MT$ is movement time, and $a$ and $b$ are empirically derived via linear regression. $ID$ is the index of difficulty, the overall selection difficulty, based on $A$, the amplitude (i.e., distance) between targets, and $W$, the target width. As seen in Equation (3), increasing $A$ or decreasing $W$ increases $ID$, yielding a harder task.

Throughput is recommended by the *ISO 9241-9* standard as a primary metric for pointing device comparison, rather than movement time or error rate alone. Throughput incorporates speed and accuracy into a single score and is unaffected by speed-accuracy tradeoffs [46]. In contrast, movement speed and accuracy vary due to participant differences. Throughput thus gives a more realistic idea of overall user performance than movement time or error rate. Our study employs throughput for consistency with other studies [35, 66, 67]. Throughput is given as:

$$TP = \frac{ID_e}{MT} \quad where \quad ID_e = log_2\left(\frac{A_e}{w_e} + 1\right) \quad (4)$$

$ID_e$ is the effective index of difficulty and gives difficulty of the task users *actually performed*, rather than that they were *presented with*. Effective amplitude, $A_e$, is the mean movement distance between targets for a particular condition. Effective width, $W_e$ is:

$$W_e = 4.133 \cdot SD_x \quad (5)$$

Where $SD_x$ is the standard deviation of selection endpoints projected onto the vector between the two targets (i.e., the task axis). It incorporates the variability in selection coordinates and is multiplied by 4.133, yielding $\pm 2.066$ standard deviations from the mean. This effectively resizes targets so that 96% of selections hit the target, normalizing experimental error rate to 4%, facilitating comparison between studies with varying error rates [45, 59].

Both visual and motor scale have been previously studied by HCI scholars in non-VR contexts, often using Fitts' law studies Factors involved in evaluating scale include the physical dimension of the device screen, the pixel density of the screen, and the distance between the user and the display screen [2, 12, 17, 33, 41, 70]. Browning et al. found that physical screen dimensions affected target acquisition performance negatively, especially for smaller screens [12]. Chapuis et al. also report that target acquisition for small targets suffered, indicating that selection performance is affected by movement scale, rather than visual scale [17]. Accot et al. used identical display conditions with varying input scale to isolate movement scaling to adjust the trackpad size systematically [2]. They used this set up with the steering task [3] and found a "U-shaped" performance curve, meaning that small and large trackpad sizes had the worst performance. They concluded that this was a result of human motor precision [2]. Kovacs et al. studied screen size independent of motor precision. Their findings suggested that human movement planning ability is affected by screen size [41]. Hourcade et al. showed that increasing the distance between the user and the screen, which causes the targets to scale due to perspective, affects accuracy and speed negatively as well [33].

## 3 WARPED VIRTUAL SURFACES

With WVS, users perceive themselves interacting with an arbitrarily sized virtual surface, that is potentially much larger than the physical tablet. The actual interaction space is always the same (i.e., the physical tracking area of the tablet). We rescale the plane representing a virtual screen in VR and render targets on locations that would fall outside the bounds of the physical tablet's tracking area. In other words, with WVS, users can select and "feel" targets that are beyond the extents of the tablet's physical dimensions.

Tablet drivers typically provide the *stylus tip* position on the tablet relative to the top or bottom left corner (i.e., the coordinate

origin of the tracking area) with $x$ and $y$ values ranging from 0 to 1. In our case, the origin was the bottom left corner. The coordinate range is calculated based on the physical distance of the *stylus tip* to the origin and dividing the $x$ and $y$ values of that distance vector by the respective physical *width* and *height* of the tablet's active tracking area. We calculate this as the *real cursor position* ($C_{Rp}$), the point where the *stylus tip* is physically touching the tablet:

$$C_{Rp} = (1/width, 1/height) \times dist(stylusTip, W_o) \quad (1)$$

Similar to haptic retargeting, we use a warping origin ($W_O$) [6] for scaling. For WVS, the origin is the centre of the physical tablet's rectangular tracking area. We chose the centre of the tablet as the warping origin because it was the point from around which the physical panel would grow in size. $W_O$ is also the only point on the tablet surface, which, regardless of the SF, remains in its original 1-to-1 mapping position. In contrast, the virtual tablet corner points are subject to scaling. Therefore, we instead chose $W_O$ as the origin point for both $C_{Rp}$ and the virtual *warped cursor position* ($C_{Wp}$), the position of the cursor the user sees on the screen panel in VR. We thus shift the coordinate system origin of $C_{Rp}$ from the bottom left corner of the tablet to $W_O$ by rescaling the output range of the $C_{Rp}$ points to range from -0.5 to 0.5 instead of 0 to 1. This results in the centre of the tablet tracking surface to be represented as (0, 0) instead of (0.5, 0.5). We track the *stylus tip* with the tablet's built-in digitizer and apply the SF only when the stylus is within tracking range, ensuring that warping is limited to the tablet's surface.

At $Wo$, $C_{Rp}$ and $C_{Wp}$ align. Warping the tablet's surface causes $C_{Wp}$ to move ahead of $C_{Rp}$ as they move the stylus further away from $W_O$. This is similar to the effects of CD gain, where a small movement of the physical mouse translates to a large screen movement for the mouse cursor. We use a similar idea to extend cursor reach on the tablet in VR with WVS. A larger SF would cause the $C_{Wp}$ to speed up, much like a high CD gain. The further we move away from $W_O$ decoupling between $C_{Rp}$ and $C_{Wp}$ increases. See Figure 1. $C_{Rp}$ values range from -0.5 to 0.5, multiplied by a *ScaleFactor* yield $C_{Wp}$, which is where we render the cursor in VR.

$$C_{WP} = ScaleFactor \times C_{Rp} \quad (2)$$

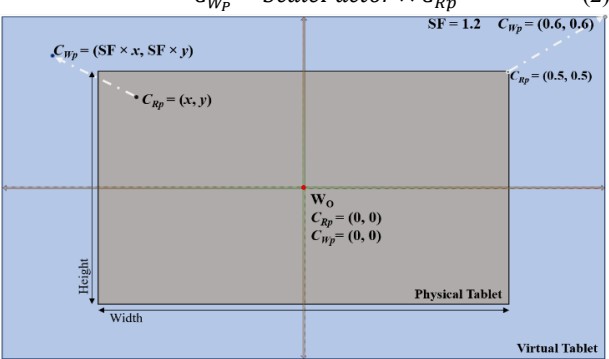

Figure 1: Visual representation of the WVS system.

## 4 METHODOLOGY

We conducted a Fitts' law experiment comparing several SFs to a 1-to-1 mapping "control condition." Our objective was to determine whether or not the application of scaling in our warped virtual surface technique influenced user performance. We used a set of pre-selected amplitude and width pairs, rather than fully crossing a selection of amplitude and widths. This ensured that all combinations of $A$ and $W$ were reachable with the 1-to-1 mapping condition. Also, a sufficiently large $W$ could yield targets that were cut off the virtual tablet screen, which would not be reachable by the cursor without warping. We thus carefully chose our amplitude and width pairs so that they would cover a wide range of $IDs$, while still having physically reachable targets under 1-to-1 mapping.

### 4.1 Hypothesis

Past studies suggest that although visual and motor scale affect performance differently, small scales and sizes impact user performance negatively for both [10, 12, 17, 20, 33, 48, 70]. Most similar to our work, Blanch et al.'s study revealed that pointing task performance is governed by motor space rather than visual [10].

Thus, we hypothesize that movement time (*MT*), error rate, target entry count and most importantly, throughput (*TP*) would be unaffected with varying SFs since participants are still selecting the same physical locations on the tablet's surface. In other words, we hypothesize that selection performance will be the same regardless of the influence of warping. We show this by using a non-inferiority statistical analysis [58] (explained in Section 5.1).

### 4.2 Participants

We recruited 24 participants (11 females, aged 19 to 64, *μ*= 26.5, *SD*=10.5). Three were left-handed, and one was ambidextrous but chose to complete the experiment using their right hand. We also surveyed their experience with VR and games: 62.5% reported having *no VR experience at all,* 37.50% reported having *a little* VR experience, and 4.20% *a moderate amount*. In terms of gaming experience, 37.50% reported having *no* 3D first-person game experience, 20.80% reported having *a little,* 29.20% reported having *a moderate amount,* 12.50% reported having *a lot of* experience. All participants had normal or corrected-to-normal stereo vision, assessed based on questioning before entering VR.

### 4.3 Apparatus

#### 4.3.1 Hardware

We used a PC with an Intel *Core i7* processor PC with an NVIDIA *Geforce GTX* 1080 graphics card. We used the HTC *Vive* VR platform, which includes an HMD with 1080 × 1200 pixel (per eye) resolution, 90 Hz Refresh Rate, and 110° field of view. The tablet was an XP-PEN *STAR 06* wireless drawing tablet. Its dimensions were 354 mm × 220 mm × 9.9 mm with a 254 mm × 152.4 mm active area, a 5080 LPI resolution. The tablet includes a stylus with a barrel button and a tip switch to support activation upon pressing it against the tablet surface. The 2D location of the stylus tip is tracked along its surface by the built-in electromagnetic digitizer. We affixed a Vive tracker to the top-right corner of the tablet using Velcro tape, see Figure 2.

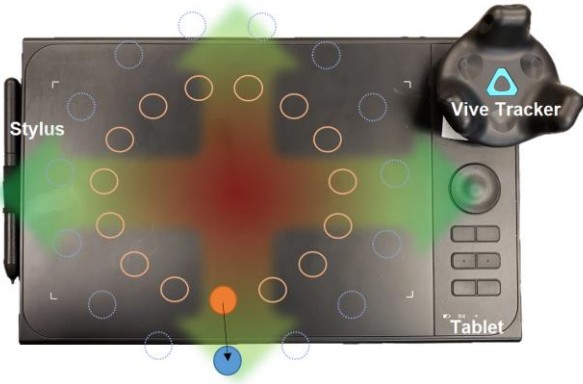

Figure 2: Overlay of Fitts' law task on the tablet. Orange circles depict the physical target location, while blue circles depict the targets the user saw in VR. Gradient arrows illustrate the surface warping effect and how the virtual surface grows in size in all directions.

#### 4.3.2 Software

We developed our software using Unity3D 2019.2 and C# on MS Windows 10. Figure 3 depicts the participants' view in VR during a selection task. We also included several space-themed assets from the Unity store. These were not seen during the task but were visible during breaks to help entertain participants between trials. We used a modified version of the source code provided by Hansen et al. [29] to develop our software.

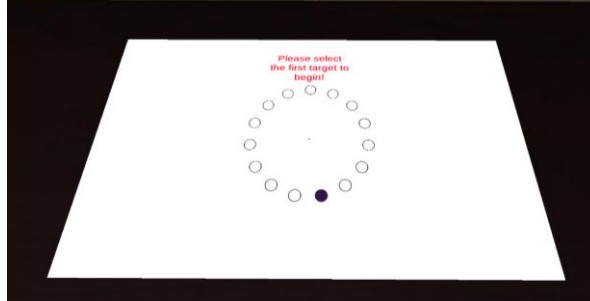

Figure 3: Fitts law task as seen in on the tablet in VR.

Windows 10 recognizes the tablet as a human interaction device profile, which causes tablet input to be mapped to the mouse cursor. We used a custom LibUSB driver that allowed direct access to the tablet data to use in Unity. The library provides stylus coordinates on the tablet surface and whether the stylus was touching the tablet surface or hovering above it within an approximately 1 cm range.

The software polled the Vive tracker to map a virtual interaction panel to the physical tablet's active tracking area, co-locating their centres. The virtual panel in VR had a resolution of 4000 × 2400, with the same size in 1-to-1 mapping as the physical tracking area on the tablet. When scaling was applied, the virtual tablet panel size was multiplied by the SF value. The tablet stylus was used to interact with the tablet. We could not find a reliable and suitable solution to track the stylus or hands externally, hence tracking was limited to the tip of the stylus in a close range to the tablet surface by the tablet's digitizer. Due to this limitation, we did not render a model of the stylus or hands. However, when the stylus was in the range of the tablet, we displayed a virtual star-shaped cursor with a dot hotspot in the centre at the stylus tip. By applying pressure and touching the tablet with the stylus, input ("click") events were detected on the tablet. The virtual cursor was used for selection, and its position was calculated as described in Section 3.

The virtual tablet sat on a table (see Figure 4). While hovering on targets, they changed colour to show which would be selected if the tip switch was pressed. Upon selecting a target successfully, an auditory "click" sound was played, and the experiment would move to the next target for selection. In case of an error, a distinct "beep" sound was used to indicate selection error, and the experiment would move to the next target in the current sequence.

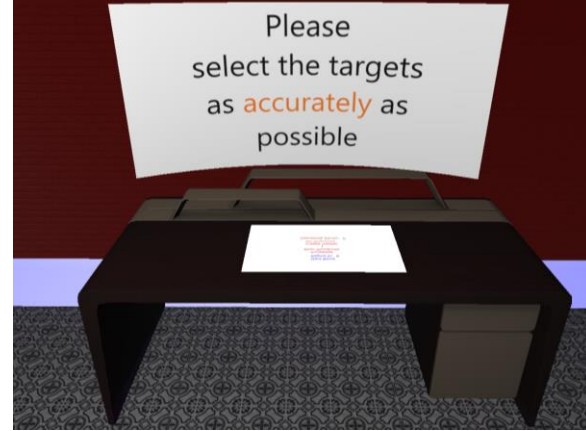

Figure 4. The virtual table where participants sat during the study.

## 4.4 Procedure

Overall, the experiment took about one hour, with participants in VR for around 45 minutes. Before starting, participants provided informed consent and completed a demographic questionnaire. The main experiment was divided up into eight blocks (one per SF). Each block consisted of ten sequences, one for each of the 10 *IDs*. In each sequence, participants were presented with 15 targets. Participants had to successfully select at least 50% of the targets to move on to the next sequence. Participants were given an (at least) 30-second break between each block. During the break, they could remove the HMD if desired. To begin each sequence, participants had to select the first target to begin the timer. This selection in each sequence was thus not logged as it just started the sequence. The target would be selected again as the last target in the sequence.

There was no training session before starting the experiment. The task involved selecting circular targets, as is commonly used in Fitts' law experiments (see Figure 2). The task required selecting the purple target as quickly and accurately as possible. If participants missed the target, the system would record an error and move on to the next target. If the participants had more than 50% error rate, they were asked to redo the sequence, the data logging system would record this. After completing the experiment, the participants exited the VE and completed a post-questionnaire where they gave comments on their experience using the VR tablet prototype. They were then debriefed and compensated $10 CAD.

## 4.5 Design

Our experiment employed a within-subjects design with two independent variables *Scale Factor* and *ID* (index of difficulty):

**Scale Factor (SF):** 1, 1.2, 1.4, 1.6, 1.8, 2.0, 2.2, 2.4.

**ID:** 1.1, 1.5, 1.8, 2.1, 2.3, 2.7, 3.5, 3.8, 4.0, 4.6.

The *ID* values were generated from the following 10 combinations of *A* and *W* (in pixels):

| ID | 1.1 | 1.5 | 1.8 | 2.1 | 2.3 | 2.7 | 3.5 | 3.8 | 4.0 | 4.6 |
|---|---|---|---|---|---|---|---|---|---|---|
| A | 300 | 450 | 1300 | 1600 | 800 | 1100 | 1000 | 2000 | 2250 | 2300 |
| W | 250 | 250 | 500 | 500 | 200 | 200 | 100 | 150 | 150 | 100 |

The SFs were applied to both the cursor position and virtual panel size in VR, as described in Section 3. *IDs* were calculated according to Equation (3) using the SF of 1 (i.e., 1-to-1 mapping). SF ordering was counterbalanced via a balanced Latin squared. Within each SF, *ID* order was randomized, with one *ID* per sequence (i.e., circle) of 15 targets.

Our dependent variables included:
- **Movement time**: average selection time, in milliseconds.
- **Error rate**: average proportion of targets missed (percentage).
- **Throughput** (in bits per second, bps): calculated based on the ISO 9241-9 standard, using Equation (4).
- **Target entry count**: number of times the cursor entered a target before selection; representative of control problems [47].

Like others [22, 45, 46, 59, 67], we argue that throughput gives a better idea of selection performance than either movement time or error rate. The accuracy adjustment used to derive throughput incorporates speed and accuracy together, making throughput constant regardless of participant biases towards speed or accuracy. It is thus better facilitates comparison between studies and is more representative of performance than speed or accuracy alone [46]. We use it as our primary dependent variable, similar to other studies.

In total, each participant completed 8 SFs × 10 IDs × 15 trials (individual selections) for 1200 selections. Our analysis is based on 24 participants × 1200 trials = 28800 selections in total.

## 5 RESULTS

We used repeated-measures ANOVA on movement time, error rate, throughput, and target entries to detect significant differences due to SF. We did not analyze *ID*, as it is expected to yield performance differences. As detailed below, we found significant main effects for SF *only* for *MT* and target entries. We did not find significant differences in error rate, and most importantly, for throughput. Horizontal bars (●••••●) indicate pairwise significant differences between conditions with Bonferonni adjustments.

We note here that while standard null-hypothesis statistical testing will determine if two conditions are significantly *different*, our objective was to determine if WVS is *not worse* than the one-to-one mapping (i.e., SF of 1). This would suggest that it has minimal impact on user performance and is thus a viable technique for virtually extending tablet surfaces. However, standard null-hypothesis statistical tests (e.g., ANOVA) do not determine if two conditions are statistically *the same* or *non-inferior* compared to one another. Hence, we instead conducted non-inferiority testing for *TP* and error rate [58].

## 5.1 Non-inferiority Statistical Analysis

Non-inferiority testing is a form of equivalence testing that shows if a condition is statistically *no worse* than another. It requires defining an indifference zone, i.e., the maximum allowed difference between two conditions to be considered non-inferior based on the context of the study [58]. With the indifference zone defined, we next analyze the mean difference between the conditions and the 1-tailed 95% confidence interval of that difference. Finally, we check if the mean difference score *and* the 1-tailed 95% confidence interval fall within the extents of the indifference zone. If so, then the two conditions are deemed to be no worse than each other, i.e., equivalent [58]. Although this form of analysis is rare in HCI, it has been used before in the context of VR Fitts' law experiments [39].

For throughput, we used the same indifference zone (1bps) as Kohli et al. [39]. For error rate, they used the smallest unit of error, i.e., one target miss in a sequence. We used the same threshold, in our case, one miss in fifteen targets, for a 6.66% indifference zone.

## 5.2 Throughput

RM-ANOVA on throughput, revealed no significant difference for scale factor ($F_{4.21, 96.98} = .92$, *ns*). Mean *TP* was fairly consistent across all scale factors. See Figure 5.

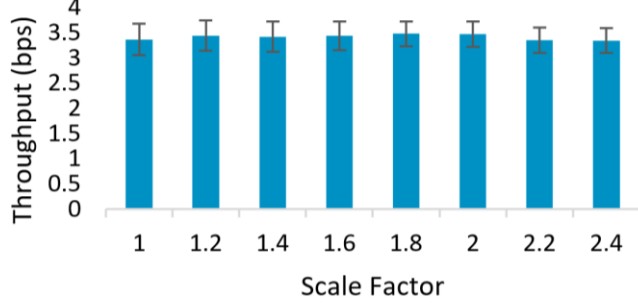

Figure 5: Mean TP for each SF value. Error bars show 95% *CI*.

To determine if throughput is statistically consistent across SFs, we next conducted a non-inferiority test. Using the aforementioned indifference zone of 1 bps, the mean difference between each compared SF and the lower bound of the one-tailed confidence intervals should be greater than -1bps to be considered non-inferior. Table 3 shows the results of the non-inferiority test for pair-wise comparisons (with Bonferonni corrections) between the 1-to-1 mapping and all other SFs. Based on this analysis, no SF has worse *TP* than 1-to-1 mapping (i.e., they are all considered non-inferior). Overall, this result indicates that *TP* is not affected by SF, in line with our main hypothesis and suggesting our WVS technique has minimal impact on user performance.

Table 3: Mean *TP* differences and non-inferiority test results.

| SF Pairs | Mean Diff. | 1-tailed 95% CI | SD Error | Non-inferiority Comparison |
|---|---|---|---|---|
| 1-1.2 | -0.077 | > -0.313 | 0.067 | -0.313 > -1.0 |
| 1-1.4 | -0.059 | > -0.333 | 0.078 | -0.333 > -1.0 |
| 1-1.6 | -0.073 | > -0.426 | 0.100 | -0.426 > -1.0 |
| 1-1.8 | -0.115 | > -0.393 | 0.079 | -0.393 > -1.0 |
| 1-2.0 | -0.108 | > -0.410 | 0.086 | -0.410 > -1.0 |
| 1-2.2 | 0.0130 | > -0.260 | 0.077 | -0.260 > -1.0 |
| 1-2.4 | 0.0190 | > -0.194 | 0.060 | -0.194 > -1.0 |

## 5.3 Error Rate

We found no significant difference in error rate for different SFs using RM ANOVA ($F_{3.98, 91.56} = 2.07$, $p > .05$). Based on Figure 6, they too are reasonably consistent across the eight scale factors.

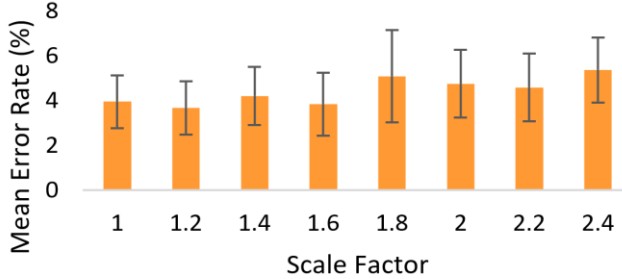

Figure 6: Error rate for each SF condition. Error bars show 95% *CE*.

Like with TP, we used a non-inferiority test on error rate to determine if each SF yielded error rates no worse than 1-to-1 mapping. Using the aforementioned indifference zone limit of 6.66%, each SF must have error rate differences no higher than 6.66% compared to the SF of 1 (i.e., 1-to-1 mapping). The results of this analysis, with Bonferroni corrections, are seen in Table 4. No SF offered a worse error rate than the SF of 1. Results indicate that the error rate was also unaffected by SF value, also in line with our hypothesis, meaning target misses rates were constant regardless of SF.

Table 4: Mean error rate differences and non-inferiority results.

| SF Pairs | Mean Diff. | 1-tailed 95% CI | SD Error | Non-inferiority Comparison |
|---|---|---|---|---|
| 1-1.2 | 0.278 | < 1.708 | 0.405 | 1.708 < 6.66 |
| 1-1.4 | -0.25 | < 1.671 | 0.544 | 1.671 < 6.66 |
| 1-1.6 | 0.111 | < 2.572 | 0.697 | 2.572 < 6.66 |
| 1-1.8 | -1.139 | < 1.636 | 0.786 | 1.636 < 6.66 |
| 1-2.0 | -0.806 | < 1.213 | 0.572 | 1.213 < 6.66 |
| 1-2.2 | -0.639 | < 1.595 | 0.633 | 1.595 < 6.66 |
| 1-2.4 | -1.417 | < 0.189 | 0.455 | 0.189 < 6.66 |

## 5.4 Movement Time

RM-ANOVA resulted in significant results in the case of movement time. We also note that as suggested by Kohli et al. [39], it is not clear what a reasonable indifference zone for movement time should be.

Mauchly's test revealed that the assumption of sphericity was violated ($\chi^2(27) = 51.33$, $p = .004$) so we applied Greenhouse-Geisser correction ($\varepsilon = .56$). There was a significant main effect of scale factor on movement time ($F_{3.98, 91.65} = 13.92$, $p < .001$, $\eta_p^2 = .37$, power = 1.00 ($\alpha = .05$)). Posthoc results showing pairwise differences and mean movement times are seen in Figure 7. Results indicate higher SF values resulted in higher mean movement time, suggesting participants moved slower with higher SFs. Our hypothesis failed in the case of movement time.

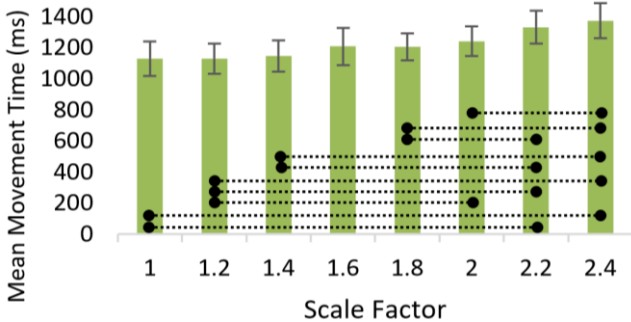

Figure 7: Mean MT for each SF. Error bars show 95% *CI*.

## 5.5 Target Entry Count

As seen in Figure 8, higher SFs yield slightly higher target entry counts, suggesting participants had more difficulty getting the cursor into the target before selection. The assumption of sphericity was not violated, so results were analyzed using RM-ANOVA as usual. There was a significant main effect of SF on target entry count ($F_{7, 161} = 17.41$, $p < .001$, $\eta_p^2 = .43$, power = 1.00 ($\alpha = .05$)). Higher SF resulted in average higher target re-entries for correct selection. Our hypothesis failed for the target entry count.

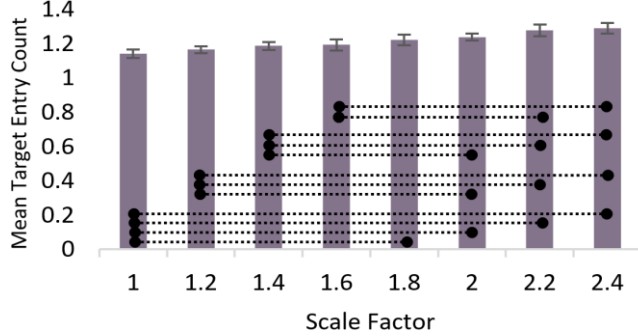

Figure 8: Mean target entry count for SFs. Error bars show 95% *CI*.

Several participants mentioned having difficulty selecting the smallest targets. As a result, we also analyzed if the target entry count was affected by the target size using RM-ANOVA. According to Mauchly's test, the assumption of sphericity was not violated. We found a significant main effect of target width on target entry count ($F_{4, 92} = 13.52$, $p < .001$, $\eta_p^2 = .37$, power = 1.00 ($\alpha = .05$)). Figure 9 depicts the mean target entry count for different target widths. Our analysis did not find a significant interaction effect between SF and target width ($F_{28, 644} = 1.33$, $p > .05$). Results suggest smaller targets were harder to hit upon initial entry and required on average more re-entries to select.

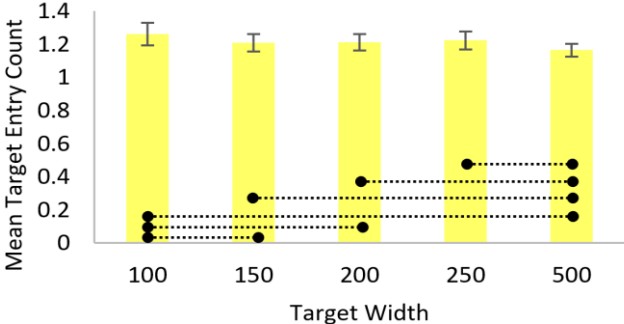

Figure 9: Mean target entry count across the target width. Error bars show 95% *CI*.

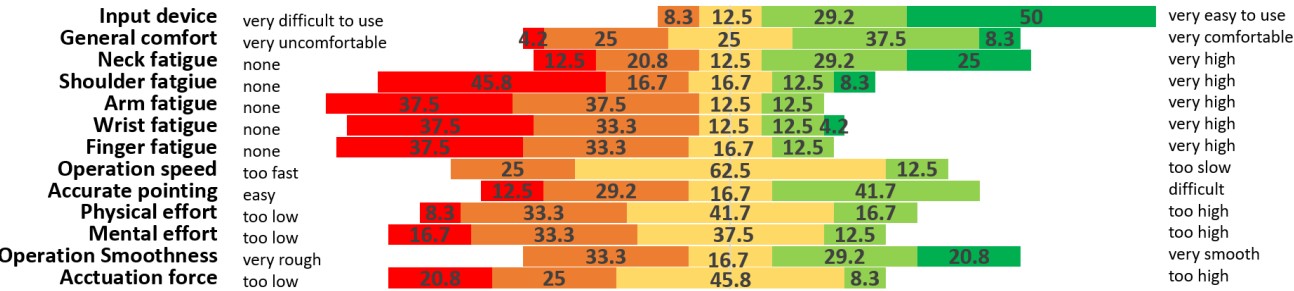

Figure 10: Device Assessment Questionnaire results. Label numbers indicate the percentage of participants choosing each answer.

## 5.6 Fitts' Law Analysis

Fitts' law is commonly used as a predictive model of movement time. We performed a linear regression of *MT* onto *ID*. Figure 11 depicts the relationship between *MT* and the *ID*.

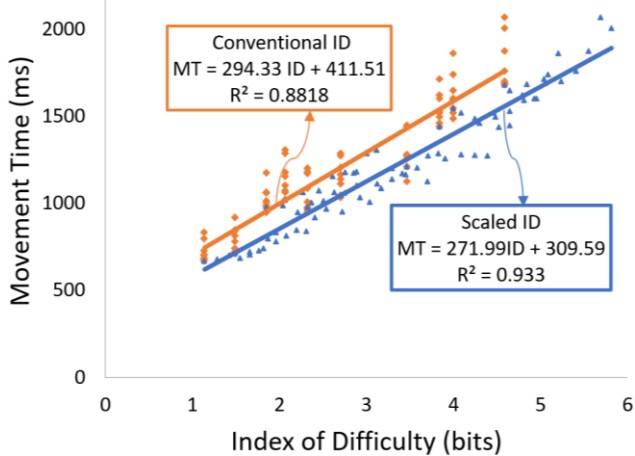

Figure 11: Linear regression of *MT* on *ID* for both presented *ID* and scaled *ID* (applying the scale factor to *A* when calculating *ID*).

As is often the case in Fitts' law studies, there is a strong linear relationship between *MT* and *ID*. We performed linear regression for both conventional *ID* (i.e., the 10 IDs listed in Section 4.5), and for scaled *ID*. Scaled *ID* was calculated based on the SF value applied to target amplitudes when calculating *ID*. Applying the scale factor in this way is more representative of the task participants *perceived* themselves as performing. Interestingly, the Fitts' law regression using scaled *ID* yielded a better fitting model.

## 5.7 Effective Width Analysis

To further explore why throughput was constant, despite increasing movement time across scale factors, we also analyze effective width. We note that throughput is based on effective width, which in turn relates to the *magnitude* of errors, rather than the error rate. This explains how throughput can stay constant across SFs while why movement time significantly increases with SF (and while error rate is also constant). Misses farther from the target will push effective width upward, while selections closer to the target centre will lower it. Thus, we looked into how mean $W_e$ changed under different SF conditions. Based on Figure 12, $W_e$ appears to decrease with higher SFs. This indicates that participants were making more accurate selections with higher SFs, likely yielding the higher movement times noted above with higher SFs.

## 5.8 Post-Questionnaire

We used the device assessment questionnaire from ISO 9241-9 [22] to evaluate the experience of using a tablet and stylus with WVS.

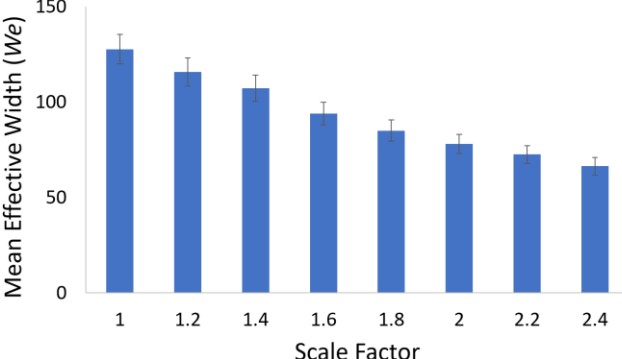

Figure 12: $W_e$ for each SF. Error bars show 95% *CI*.

Participants had to rate each phrase from 1 (lowest) to 5 (highest).See Figure 10. We did not compare results across different SFs since participants were unlikely to notice differences in the SFs, and this would require they complete 8 lengthy questionnaires instead of just one.

## 6 DISCUSSION

Results indicate that WVS had significant effects only on *MT* and target entry count. Notably, the largest SFs were significantly different. For error rate and *TP*, non-inferiority testing indicated that WVS is no worse than 1-to-1 mapping. Target entry count was affected by the target size, which was unsurprising, but highlights some difficulty in accurately selecting the smallest targets.

### 6.1 General Discussion

Our results are in line with past work and suffer from some of the same limitations [10, 39]. The most important result from our study is that throughput is relatively stable when using WVS, especially for modest scale factors, e.g., 1.2, 1.4 and 1.6. One explanation could be that the same muscles are used across all selections under different SFs. Pointing performance is known to be affected by the muscle used to reach the target [74]. This is a promising finding, as it suggests that WVS can be applied in tablet-based VR to provide a larger virtual tactile proxy than is otherwise available. Similarly, as indicated in our results, movement time is significantly (if only slightly) worse with higher SFs, especially at 2.2 and 2.4.

Overall, our results show lower throughput with a tracked stylus compared to redirected touching [39]. Lower throughput and error rate and higher movement time in our study are potentially due to the different warping techniques we used. Other factors that could contribute to this difference are the different hardware setup, for instance, input using a stylus rather than fingers, and the position/orientation of the tablet in our study. Notably, our throughput scores – regardless of scale factor – are in line with previous work using a 3D tracked stylus, which is a closer comparison point anyway [67]. Also, the tablet placed on the table

caused participants to experience some neck fatigue, as indicated in the post-questionnaire results (see Figure 10) and participant comments. 54.2% of the participants reported high neck fatigue. Our participants also noted it was hard for them to select the smaller targets. One commented, "*There were some times where selecting the smaller circles was difficult.*" Such comments were not unexpected and are supported by the significant differences found in our analysis, as shown in Figure 9. One other contributor to this difficulty could be the limited screen resolution in the Vive HMD.

Higher scale factors yielded slightly higher movement times and target entries than lower scale factors. A potential reason for this is the increase in virtual cursor movement speed caused by scaling. This increase in cursor speed would make fast, accurate movement more challenging, particularly in precisely selecting targets. This kind of effect has been noted before as a "U-shaped" curve for coarse/fine positioning times under different CD gain levels [2]. Also, since users were not able to see the stylus in VR, they likely moved more slowly to keep track of the cursor.

On the other hand, as seen in Figure 5, throughput is almost flat across scale factors. Throughput characterizes the speed/accuracy trade-off in selection tasks. For throughput to be flat across scale factor, and in light of increasing movement times, accuracy must have been better with higher scale factors. In our error rate analysis, we found non-inferiority between 1-to-1 mapping and all other scale factors, suggesting error rates were at least *not worse* with higher scale factors. However, effective width (from which throughput is derived) is not based on error rate, but rather on the distribution of selection coordinates. In other words, the distance of the selection coordinates to the target centre influences $W_e$. Participants may miss targets at about the same rate, but miss "closer" to the target (which yields lower $W_e$). Alternatively, they may hit closer to the centre of the target (which also yields lower $W_e$). This is confirmed in our $W_e$ analysis (Section 5.7) – $W_e$ became smaller with higher scale factors, which is why throughput was constant regardless of the SF. With higher SF, the cursor moved faster. Participants likely slowed their operation speed slightly to compensate for the higher cursor speed. This is reflected in higher $MT$ for higher scale factors (Figure 7). By compensating (i.e., slowing down), participants were more readily able to precisely select targets (yielding lower magnitude misses, or selections closer to the target centre), resulting in lower $W_e$ and higher throughput.

Based on our observations, most target misses were due to loss of tracking for the stylus and participant moving their hand closer to regain tracking and accidentally touching the tracking surface. Some participants also commented on this. A participant reported: "*my errors were false selection during dragging my hand to the desired point.*" Also, since we were warping the virtual space, moving the stylus even slightly could cause the virtual cursor to move outside smaller targets (increasing target entry count). Participants held the stylus at an acute angle relative to the tablet surface, instead of perpendicular to it; reaching and selecting smaller targets from the hovering state could cause the virtual warped cursor to fall outside the target even with slight movements in either direction. Participants also held the stylus differently, i.e., in a different position and with different gestures.

Participants found WVS easy to use, despite 41.7% finding accurate pointing *difficult*. One participant reported: "*Overall was easy to select the targets.*" Another participant mentioned that "*...would use again. Was more usable when the in-world representation of the tablet was larger, but it was still easy to select small targets on the small display.*" Half of our 24 participants reported the device to be *very easy to use* (Figure 10).

Based on comments, participants liked that they could use a larger touch surface in VR despite arm, wrist and finger fatigue, as indicated in Figure 10. One participant commented that "*I really like the idea of using smaller physical screens to choose on larger area in VR, hope it will become common input option for VR.*" Only three out of 24 participants reported they did not notice any change in their cursor movement speed while in VR. One person mentioned that "*I was able to observe the warping but not able to compare it to earlier trials. It was a smooth experience.*"

## 6.2 Limitations

The main limitation of our study is the indifference zones used for the non-inferiority analysis. More studies are required to determine valid indifference zones for performance in Fitts' law studies. In the presented work, we used the same indifference zones as Kohli et al. [39] for the sake of consistency and to facilitate comparison. As mentioned in their work, some previous studies have found significant differences between conditions within the chosen indifference zones. Although we demonstrated non-inferiority, different indifference zones or statistical analysis could yield different results. Another limitation is that our hardware setup did not support 6DOF stylus tracking. We believe our findings can still be useful and can help VR researchers and system designers.

## 7 CONCLUSIONS

We introduced Warped Virtual Surfaces, a technique to scale input space with a tracked tablet, yielding larger virtual tablets than that physically available. We evaluated the effects of surface warping on task performance using a tablet and stylus in VR. In terms of *TP* and error rate, WVS yielded consistent performance regardless of SF. Non-inferiority statistical tests showed that *TP* and error rate were statistically similar between all tested SFs and the "control" condition, i.e., the 1-to-1 mapping. However, for movement time and target entry count, we found small but significant differences, particularly for larger SFs, in line with previous work [10, 39].

Our proposed method can be used for artists and designer that are interested in immersive workflows or for VR design sessions. Our approach uses cheap and affordable hardware. It enables users with a fix-sized physical panel or drawing tablet to get a bigger virtual panel without extra hardware and performance cost. WVS could be useful with small, lightweight arm-mounted touchscreens to facilitate tactile interaction with 3D menus or similar applications to PIP and WIM [61, 63]. WVS can also complement other tablet and stylus-based interaction techniques for VR, such as the HARP system [44], the Virtual Notepad [53]. In-Air drawing applications could also benefit from WVS. Interaction techniques like snappable panels or surfaces could employ WVS. WVS could potentially help with fatigue, but further experiments are needed to determine to what extent. Other haptic devices with limited interaction space, like the Phantom, could potentially also benefit from WVS by expanding their virtual reach.

We conclude that our technique shows promise as a method to virtually extend physical surfaces in VR. Results suggest minimal performance impact of WVS. *TP* was flat across all SFs. Despite small differences in *MT*, it seems users made up their performance via a slight accuracy improvement, yielding constant *TP*.

Future work on Warped Virtual Surfaces will involve a follow-up study across multiple scale factors and multiple tablet sizes. We will use a subset of scale factors from the current study, with physically smaller tablets than that used in this study. We will also employ a 3D tracked stylus (e.g., Logitech VR Ink) scaling 3D movement, rather than just planar movement.

### ACKNOWLEDGEMENTS

We would like to thank Kyle Johnsen and A. J. Tuttle for sharing their tablet driver source code. We also thank our participants and other researchers whom their work inspired this research. This research was supported by NSERC.

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
