# OpenReview forum: "Selection Performance Using a Scaled Virtual Stylus Cursor in VR"
_graphicsinterface.org/Graphics_Interface/2020/Conference — GI 2020_

### Official Review · AnonReviewer3 · 2020-04-16
**Good paper with an interesting technique but needs some organization and clarity work**

**Rating:** 6
**Confidence:** 5

**Review:**

This submission presents a new VR-based interaction technique, warped virtual surfaces. With this technique, the user holds a stylus and moves it along the surface of a tablet, however, instead of the stylus having a 1:1 mapping with it’s movement along the table surface, the mapping is warped to allow the user to interact with objects that are much further away / in a larger planar interaction volume. This technique was evaluated via a 24-person user study using a Fitts’ Law reciprocal tapping task. The results demonstrated that regardless of the scale factor, throughput and error rate were consistent between the new technique and the use of a traditional 1:1 mapping technique.

This submission is quite well written. The technique is interesting and I can see how it could be useful for some VR-based tasks. It would be useful to talk a bit more what applications the technique may apply to and how the author(s) think that the technique could extend to non-table-based interaction (the tablet was on the table in this study, but this likely won't be the case for many applications in the future).

I found the related work section to be through and the included references seem complete. However, the related work would benefit from a summary paragraph (either at the end of each subsection or the end of the entire section) that delineates how the experiment / technique / findings from this work differ from the cited work. In Section 2.2, for example, I was left wondering  what was important and relevant about the JND and 2AFC methodologies that were mentioned and how the results from these studies related to the design of this study or the proposed technique (note that these are methodologies so the word ‘methodology’ should be included in the text and the JND result summary needs to explain what the referent level was). Ditto for Sections 2.1 and 2.3. Section 2.4 also seems unnecessary given that Fitts’ Law is well known within the HCI literature and I suspect the readership of this paper is already familiar with the derivation of throughput. Section 2.4 can probably be removed.

In terms of the presented technique itself and the methodology used in the study, I found the explanations to be clear and easy to follow. The discussion noted that not all participants held the stylus the same way, although they all held it at an angle, and in some instances small movements of the stylus led the cursor to fall outside a target.  This comment led me to wonder about the calibration and accuracy of the tablet – specifically where the electromagnetic receiver was located in the stylus (i.e., right at the tip of the nib or was it further up in the barrel) and the role that this had on the accuracy of the pointing (i.e., the further away the receiver is from the nib, the more of an influence holding the stylus at an angle will have on the detected tip position, and thus the cursor and warped positions). It would be useful to include a discussion on this and detail how the tablet was calibrated (i.e., with the stylus perpendicular to the tablet or at an angle).

One challenge I have with this paper is that there are a lot of metrics that are reported and graphs / figures that are shown without a clear message. Because of this, it is very easy to get bogged down in trying to understand what the results of each metric actually mean and how various results related to each other and what the overall findings of this work are. To make it clearer why each metric is important, it would be a good idea to remove the metric discussion from 4.3.2 Software and create a specific metric section that justifies each metric and highlights the hypotheses of the experiment. Moving this statement from the Discussion up into a metrics section would also help the reader while they parse through all of the findings in Section 5: “Similar to other authors [22, 44, 45, 57, 64], we argue that throughput gives a better idea of overall selection performance than either movement time or error rate. This is because the Accuracy adjustment Used to derive throughput incorporates speed and accuracy together, making throughput constant regardless of participant biases towards speed or accuracy. It is thus a better point of comparison to other studies, and more representative of performance than speed or accuracy alone [45]. ”

While the discussion does try to tie the results together, it is very confusing, especially when there are statements that refer back to Figures but the statements don’t actually detail what about the figure is important or participant comments are mentioned without the actual quoted text. For example, “Also, the tablet placed on the table caused participants to experience some neck fatigue, as indicated in the post-questionnaire results (see Figure 12) and participant comments.” This statement doesn’t actually refer to what the results were, so I have to go back to the Figure. Because there is no synthesis of the results in the text in Section 5 and no quotes are given, the reader is left to make their own guesses about what the author(s) intend. To fix this, I recommend adding a summary sentence at the end of each subsection in Section 5 to identify what each result means at a high level (e.g., “So these throughput results suggest that the WVS technique performs no worse than the use of traditional 1:1 mappings”) . Adding in participants comments would be beneficial - especially any comments about the haptic feedback that was given by the hand and stylus on the tablet during interaction (which the paper touts as the motivation for using this technique to begin with). It would also be useful to include more sub-sectioning in Section 6 and moving Section 7.2 into Section 6 so that all the limitations can be discussed together (i.e., paragraph 2 of Section 6 already discusses some limitations).

Overall, the bones of this paper seem to be quite good, however, the organization and clarity of the conclusions need quite a bit of work. Fixing the related work, applications, and stylus accuracy is quite easy, but I am on the fence as to how much transformation is needed within Section 5 and 6 (which really form the meat of the paper) to make the paper easier to digest and the findings clearer. Because of this, I am assigning a marginal rating.

---

### Official Review · AnonReviewer1 · 2020-04-18
**Missing use cases, but good study execution**

**Rating:** 6
**Confidence:** 3

**Review:**

#== SUMMARY ==#

The authors present a warping technique for tablet-based VR input. With their technique, the CD ratio of pen input changes the further away the user moves the pen from the tablet's center. The main part of the paper is a Fitts' Law test in which participants had to select virtual targets as quickly and accurately as possible. The authors conducted statistical significance tests like ANOVA. In addition, they used non-inferiority tests to find out whether a warped CD ratio can be seen as equal in user performance compared to a constant CD ratio. The authors found that while movement time increases when using a warped CD ratio, the throughput can be seen as statistically equal between conditions.

#== REVIEW ==#

The presented idea is nice and simple. The authors support it well based on related work and position their work quite clearly. The study is well motivated and executed. The system implementation is well described, which improves reproducability.

Unfortunately, it is hard to clearly see what to learn from the paper. While the results provide a good indication that users would be able to effectively interact in such a warped interface, the findings are not overly surprising and the technique itself is not very fleshed out.
One problem I see is the lack of a strong use case to support the study design. There are many use cases for general warping and redirecting in VR and the authors described those well. However, as opposed to examples from previous research, the authors did not provide strong and more specific use cases for their particular technique. While warping is a common theme, previous researchers in that area designed their techniques with specific use cases, applications and constraints in mind. For instance, NaviFields by Murillo et al. warp the user's locomotion based on points of interest (i.e., pre-defined regions in the virtual environment where more accuracy is desired). The Go-Go technique has the user as the warping origin to allow fine grained selection and manipulation close to the user. Other examples argue with fatigue (e.g., Feuchtner et al. as cited in the presented work).
In the presented work, such an argument or reason for warping in this particular way is lacking. More specifically, why is the center of the tablet chosen as the warping origin? Do envisioned applications require more accuracy in the center than in the periphery, i.e., is the point of interest always in the center? Is that the most ergonomic area of tablet input? The authors mainly argue with enabling interactions beyond the limits of the tablet, but this can be trivially achieved by uniformly scaling the input coordinates, i.e., right now there is no argument supporting why not only more space, but also more accuracy is needed at the origin compared to the periphery. Only then warping is necessary in the first place.
The task design in the study is then becoming questionable, because the Fitts' Law targets are arranged in a rotationally symmetric layout around the center, which is also the warping origin. As far as I understand, for each movement, the cursor is slowing down, passing through the origin and then speeding up again. There is no variation of different changes of the CD ratio across selections. While the study is still insightful, I believe that adjusting the task to the technique would have been interesting, even if it would not have been a conventional Fitts' Law task. For instance, placing the targets to random positions while adjusting their target size to the CD ratio at that point would potentially reflect the actual use of the technique better, as the path from one target to the a other would not always pass though the origin and users would need to visually adapt to different changes of the CD ratio (again, such "use" of the technique and how this study design would reflect it would then also need to be described). Furthermore, different "regions of interest" could be defined to assign different CD ratios to different regions to then, e.g., test moving from one region of interest to another. I do not insist on those particular examples, but I would have liked to see a better connection between the task and the proposed technique - possibly in context. As of now, I am unsure how to apply the findings.

In summary, there are some shortcomings. A concrete use case and/or application would have supported the proposed technique and study. However, the work is well focused and executed. A better framing can potentially mitigate the shortcomings. I therefore slightly lean towards acceptance of this work.


#== MINOR ISSUES ==#

-The figures in the paper and their layout can be improved. For instance:
	-Figure 12 should be moved to the top of the subsequent page
	-Figure 1 has a lot of empty space and does not visually summarize the paper. The depiction itself might be useful, but not as a teaser. The easiest fix would be to display the tablet and overlay in a one-column figure instead of in a teaser figure. If an additional teaser figure is used, then the content and caption should describe the paper as a whole.
-As described in the review, a strong use case or example applications are lacking. In connection to this, I did not understand why a virtual scene with various assets was created and then described in the Software section within Apparatus (4.3.2). Was that virtual environment shown to users? According to the procedure, they entirely focused on the Fitts' Law task. In addition, was that demo scene interactive or what was the general purpose of it? If this is an envisioned example application with selection and manipulation techniques (with varying levels of accuracy?) then this should be re-contextualized and described in one of the introductory sections.
-The related work is thorough with a large amount of references and I only have minor suggestions for improvements. First, I suggest to have sub section 2.4 (Fitts' Law) as a dedicated section or as a sub section in methodology. Second, a small summarizing paragraph at the end of the related work section (which would then be right after 2.3) might be beneficial to emphasize the gap in literature and transition to later sections.
-Having the limitations section as the very last sub section seemed a little bit odd to me. I suggest to add a dedicated Limitations and Future Work section before the Conclusion or to remove the sections and incorporate the contents in the discussion.

---

### Official Review · AnonReviewer2 · 2020-04-20
**Review of WWS**

**Rating:** 5
**Confidence:** 4

**Review:**

WWS is a gain-based selection method that allows the use of traditional inputs in VR. In particular, the paper proposes the use of a pen + tablet to select objects in any size of virtual panels. To evaluate this interaction technique the authors ran a traditional Fitts’ Law study, where they evaluated different scale factors and IDs. Their goal was to see if user performance when using WWS is like 1:1 mapping.

I have a couple of problems with the paper, and I will discuss them next:

1)	Previous work: The introduction and related work section discuss papers from too many different areas. I recommend the authors to remove the mention of other haptic feedback devices and redirect walking. And to focus less on the theory behind visual illusions and detection thresholds for illusions, as the paper contribution is not in this area.

2)	Previous work: I suggest the authors to better explain the differences of other gain-based selection methods and WWS, as here is where the novelty of their interaction technique relies.

3)	Study design: There is no discussion about the selected W and A, and if they might cofound the results. See [10.1145/3173574.3173770]. Regardless if this was considered or not, it is important to mention this.

4)	Study design: why was the virtual environment populated with space objects? Was there an expectation of this affecting the result?

5)	Study design: Pointing performance is affected by the muscles used to reach the target. See [10.1145/238386.238534]. However, in the used tablet, all scale factors use similar muscles. This might be one of the reasons behind the results, but there is no mention of this cofound in the discussion

6)	Results: Even if the effect of ID is well known, I think it is important to include the analysis of ID in the paper. Specially to see if there is an interaction between ID and SF.

7)	Results: The authors should calculate the movement paths. See [10.1109/MCG.2009.82]. And discuss any difference between SF. This might show interesting results that will make the paper stronger.

I think the WWS interaction technique is novel and interesting, but the paper is not ready for publishing. The introduction and related work need work, and there are a some considerations in the study design and analysis that need to be addressed.

---

### Meta-Review · Area_Chair1 · 2020-04-23

**Recommendation:** Accept
**Confidence:** 3

**Metareview:**

All reviewers identified several issues with the study. Especially R2 lists several shortcomings, some of which can be considered major. At the same time, reviewers appreciate the simple, but nice idea of the technique and acknowledge the novelty of the technique and study.

While the paper is mostly easy to follow, some of the negative points are due to a lack of clarity in the paper. R1 states that a strong use case is lacking. R1 and R2 were confused about the fact that the authors created an immersive scene and populated it with 3D content, while a descriptions of its actual use and role is lacking. R3 raises several points about the discussion and how results should be presented less confusingly. In addition, R3 has several other detailed suggestions to improve the writing, some of which overlap with the suggestions by R1. R2 suggests to focus the review of related literature more strongly.

Overall, this is a borderline paper and could go either or. However, given that the authors incorporate the writing-related feedback from the reviewers in a minor revision, this paper reaches the bar for acceptance.

---

### Decision · Program_Chairs · 2020-04-25

Accept